# The Future of Telemedicine for Obstructive Sleep Apnea Treatment: A Narrative Review

**DOI:** 10.3390/jcm13092700

**Published:** 2024-05-04

**Authors:** Sébastien Bailly, Monique Mendelson, Sébastien Baillieul, Renaud Tamisier, Jean-Louis Pépin

**Affiliations:** 1HP2 Laboratory, Inserm U1300, Grenoble Alps University, 38000 Grenoble, France; sbailly@chu-grenoble.fr (S.B.); mmendelson@chu-grenoble.fr (M.M.); sbaillieul@chu-grenoble.fr (S.B.); rtamisier@chu-grenoble.fr (R.T.); 2Laboratoire EFCR, CHU de Grenoble, CS10217, 38043 Grenoble, France

**Keywords:** telemedicine, obstructive sleep apnea, diagnosis, continuous positive airway pressure, adherence, artificial intelligence, collaborative care, virtual sleep laboratory

## Abstract

Obstructive sleep apnea is a common type of sleep-disordered breathing associated with multiple comorbidities. Nearly a billion people are estimated to have obstructive sleep apnea, which carries a substantial economic burden, but under-diagnosis is still a problem. Continuous positive airway pressure (CPAP) is the first-line treatment for OSAS. Telemedicine-based interventions (TM) have been evaluated to improve access to diagnosis, increase CPAP adherence, and contribute to easing the follow-up process, allowing healthcare facilities to provide patient-centered care. This narrative review summarizes the evidence available regarding the potential future of telemedicine in the management pathway of OSA. The potential of home sleep studies to improve OSA diagnosis and the importance of remote monitoring for tracking treatment adherence and failure and to contribute to developing patient engagement tools will be presented. Further studies are needed to explore the impact of shifting from teleconsultations to collaborative care models where patients are placed at the center of their care.

## 1. Introduction

Obstructive sleep apnea (OSA) is one of the most common chronic diseases and affects nearly one billion people worldwide [1,2]. The prevalence of OSA is expected to continue to rise due to the epidemic of obesity, physical inactivity, and diabetes, all of which are risk factors for OSA [3]. Patients with OSA present significant heterogeneity and diversity in clinical presentation and responses to treatment [4,5,6]. Moreover, OSA, in its heterogenous presentation, is related to comorbidities’ occurrence, and co-evolution and aggregation can emerge over time [7,8,9]. The first-line treatment for OSA is continuous positive airway pressure (CPAP), which opens and stabilizes the upper airways during sleep. Currently, millions of patients are treated by CPAP worldwide and CPAP treatment relies essentially on ambulatory care. CPAP is highly effective in improving patient-reported outcome measures (PROMs) and cardiovascular risk only in adherent patients. Several studies show that all-cause mortality is associated with CPAP adherence or CPAP continuation [10,11,12]. 

Traditionally, the diagnosis of OSA relies on overnight polysomnography conducted in a sleep clinic, a process often characterized by lengthy waiting lists and high demand for human resources, as well as carrying the potential for misdiagnosis and severity misclassification (i.e., due to night-to-night variability in the apnea–hypopnea index (AHI) [13,14,15,16,17,18,19,20]). Thus, among the current requirements for an accurate diagnostic workup of OSA, it is recommended that at-home multi-night testing becomes mainstream practice [21]. This situation requires the development and validation of new end-to-end digital medicine solutions supported by artificial intelligence [22].

Moreover, despite the benefits of CPAP treatment, CPAP adherence remains low. A study based on half a million patients showed that up to 50% of patients stop CPAP therapy in the first 3 years after initiation [23]. Thus, a new organization of the follow-up management pathway will be necessary in the future. Reinventing the diagnostic tools alone is a first step; however, there is a need to transform the entire patient journey from the suspicion of OSA to ongoing treatment management follow-up. 

Telemedicine is defined as the remote delivery of healthcare services, leveraging telecommunications technology to facilitate the exchange of medical information between patients and healthcare providers. Telemedicine has the potential to offer patients convenient access to healthcare [24] and offers remarkable potential to lower healthcare costs and enhance access to various care services for underserved populations [25]. Telemedicine is already used in various pathologies and it has been shown that it is an acceptable alternative to in-person care as evidenced in a recent study examining the feasibility, effectiveness, and acceptance of virtual visits as compared to in-person visits among clinical electrophysiology patients during the COVID-19 pandemic [26].

OSA treatment is well designed for telemedicine with remote telemonitoring, which is already deployed in pediatric (for review: [27]) and adult populations. In France, a new national CPAP telemonitoring and pay-for-performance scheme for homecare providers was implemented in 2018 and now involves 98% of CPAP-treated patients in France. This enables healthcare professionals to track sleep patterns and treatment adherence without the need for frequent in-person visits. The COVID-19 pandemic has led to the development of remote medical consultation systems, which have helped to advance telemedicine approaches in the context of OSA. Conversely, the COVID-19 pandemic also exacerbated existing challenges in accessing sleep laboratories, leading to significant variability in the duration of accessing diagnostic exams for OSA [28,29]. This delay in diagnosis and subsequent treatment poses limitations, particularly given the reliance on one-night sleep measures that fail to capture the full variability of the condition. However, amongst these challenges, telemedicine has emerged as a promising solution, offering out-of-lab exams that demonstrate comparable performance to traditional polysomnography (PSG) for diagnosing sleep apnea [30,31,32]. Few studies to date have investigated the cost-effectiveness of telemedicine in the management of OSA. Nevertheless, a randomized controlled trial comparing a telemedicine-based CPAP follow-up strategy compared with standard face-to-face management showed that telemedicine is cost-effective mostly due to savings on transport and less lost productivity (indirect costs) [33].

In this narrative review, we aim to investigate what could be the future of telemedicine in the treatment of OSA by exposing how home sleep studies driven by technological innovations and artificial intelligence can improve the diagnosis of OSA by decreasing delays in diagnosis and increasing access to evaluations. We will also present the importance of remote monitoring for tracking treatment adherence as well as treatment failure and the potential of remote monitoring to contribute to the development of patient engagement tools and telerehabilitation programs. Lastly, we will present the shift from teleconsultations to collaborative care models that place the patient at the center of his/her care. Lastly, we will present the pitfalls of telemedicine and the challenges remaining in the development of remote management pathways. 

## 2. Home Sleep Studies Driven by Technology Innovations and Artificial Intelligence Analysis

Traditionally, the diagnosis of OSA relies on overnight polysomnography conducted in sleep clinics, often leading to lengthy waiting periods due to high demand and the intensive need for human resources. Polysomnography is complex to implement and also requires specific expertise for analysis. Furthermore, it is widely acknowledged that a single night of polysomnography can result in the misclassification of disease severity for approximately one third of patients with mild-to-moderate sleep apnea [17]. This discrepancy stems from significant night-to-night fluctuations in the apnea–hypopnea index (AHI) and inconsistencies in sleep quality [13,14,15]. These factors include wake/sleep habits, alcohol or drug consumption, social jetlag, rostral fluid shift, natural night-to-night changes in sleep architecture (i.e., respective proportions of time spent in non-rapid eye movement [NREM] and rapid eye movement [REM] sleep), and changes in body and head positions during sleep [34,35,36,37]. 

The American Academy of Sleep Medicine has endorsed clinical practice guidelines supporting home sleep apnea testing (HSAT) using technically appropriate devices for diagnosing OSA in “uncomplicated adult patients presenting with signs and symptoms that indicate an increased risk of moderate to severe OSA” (strong recommendation) [38]. The impact of the “first night effect” is notably significant within sleep laboratories, as the unfamiliar setting can result in shifts in sleep architecture, alterations in sleeping positions, and consequent variations in respiratory event measures [15]. Thus, among the current necessities for a precise diagnosis of OSA, it is imperative that multi-night testing at home becomes standard practice [21]. Although implementing such a strategy has faced challenges, including potential increases in healthcare expenses and inconvenience for patients, the advent of new digital medical solutions promises to streamline the process of multi-night testing at home, enabling the establishment of efficient and cost-effective diagnostic pathways [17,39,40,41]. 

Home sleep studies (home sleep apnea testing or HSAT) offer a more accessible and convenient alternative to traditional in-lab sleep studies. Patients can conduct sleep studies in the comfort of their homes, which may increase overall participation in diagnostic assessments, especially among those who might find attending a sleep clinic challenging and expensive [42]. This advancement signifies a new potential avenue for expanding access to both diagnosis and treatment for OSA, addressing critical gaps in care exacerbated by the pandemic. A major step forward in the seamless integration of new diagnostics would be the validation and availability of a select set of innovative sensors and metrics capable of consolidating comprehensive information necessary to understand the pathophysiology of sleep disorders and assess disease severity [31,43]. Numerous technologies have emerged for the diagnosis of sleep apnea in the home environment, with promising techniques such as mandibular jaw movements (MJMs), photoplethysmography (PPG), and peripheral arterial tone demonstrating high performance in various research and clinical contexts [44]. Developing access to a telemedicine solution for OSA diagnosis could involve a virtual sleep laboratory, as proposed previously [22], with a view to improving (and increasing) access to adequate treatment for OSA, which remains an under-diagnosed chronic sleep disorder. Such a virtual sleep laboratory would propose preliminary screening with a recommendation for a home sleep test, if appropriate, (ideally, over several nights), data collection, and interpretation in a digital pipeline, strictly adhering to data protection rules [22,45,46]. 

## 3. Remote Monitoring, Treatment Adherence, and Treatment Failures

Once OSA is diagnosed, the first-line treatment for OSA is CPAP. Machine or web-based tracking systems that generate information for both the healthcare provider and the patient are a new aspect of CPAP devices. Over the last 20 years, telemedicine technology has been applied in the field of CPAP. This has facilitated the follow-up process and allowed healthcare providers to provide more consistent care. Despite advancements in device technology over the past two decades, CPAP termination rates remain persistently high [23], highlighting a significant adherence challenge. Interestingly, factors such as OSA severity, measured by the apnea–hypopnea index, and technical features of CPAP devices appear to have minimal impact on adherence rates. Instead, attention must be directed towards recognizing and addressing other influential factors such as comorbidities, psychological factors, relationship dynamics, socioeconomic status, access to care, and cultural diversity [47,48,49]. Tailored interventions should be developed to address these multifaceted determinants of CPAP adherence, focusing not only on enhancing device usage but also on promoting overall lifestyle changes, encompassing physical activity and dietary habits. Access to these personalized strategies can be facilitated through improved visualization tools on CPAP remote monitoring platforms and the widespread adoption of telemedicine services, incorporating innovative analytics like artificial intelligence to enhance efficacy and accessibility.

Remote monitoring platforms also offer a valuable means of detecting potential failures in positive airway pressure (PAP) treatment and facilitating timely interventions. Initially, these platforms can help pinpoint instances of elevated or fluctuating residual apnea–hypopnea index (AHI) levels under PAP therapy, as highlighted in a study by Midelet et al. [50,51]. Additionally, through telemedicine, healthcare providers can effectively identify cases of high residual AHI attributed to specific issues with PAP equipment. Treatment failure induced by mask change can be directly monitored from PAP remote monitoring data using automated algorithms which are able to automatically assess the change in rAHI level related to a mask change and produce a significant alert which can be used as a telemedicine tool [51]. Another possibility is to identify the point in time in the remote monitoring data at which an elevated residual AHI may be indicative of an exacerbation of cardiovascular comorbidities, leading to a clinical alert for the patient’s healthcare provider [52,53,54]. This implies the integration of some machine learning algorithms directly in the remote process, such as some machine learning algorithms to consider rAHI variability [55]. 

This integration of remote monitoring and telemedicine not only enhances the early detection of treatment inefficiencies but also enables targeted interventions to optimize patient outcomes and improve overall therapy adherence. The French health system has demonstrated for the first time that a national deployment is feasible and financial performance incentives encourage homecare providers to redirect resources and interactions towards individuals who have the lowest adherence to CPAP.

## 4. Patient Education and Engagement and Telerehabilitation Programs

In line with the widespread use of remote monitoring platforms, over the past few years there has been an increase in the access to online medical information, a proliferation of health apps on mobile phones and wearable devices, and a rising utilization of patient portals within electronic medical records [56,57,58]. Taken together, this clearly demonstrates that patients increasingly want to be engaged in their own health management. As mentioned previously, the ability of CPAP devices to record data on usage, efficacy, leak, and other parameters has transformed the management of patients on CPAP. Telemedicine provides a unique opportunity to collect objective data regarding adherence, efficacy, and leak data, along with patient reported outcome measures (PROMs). A recent study evaluating the impact of PROMs on CPAP usage in a real-world setting demonstrated a relationship between PROMs and CPAP use, in particular self-reported sleepiness and its response to therapy [59]. The authors of this study highlighted the potential of capturing PROMs using digital solutions during the course of treatment in order to enhance patient outcomes by providing actionable insights.

Patient engagement can be achieved through web-based access to CPAP therapy data and other asynchronous telemedicine approaches. A recent Cochrane review examined the effects of supportive, educational, and behavioral interventions on CPAP adherence. These different intervention types increase CPAP usage with varying degrees of effectiveness, which may be related to the heterogeneous nature of the factors affecting CPAP use [60]. Behavioral therapy was shown to increase machine usage by 79 min per night, and ongoing supportive interventions increase machine use by about 42 min per night. This is significantly greater than the MCID of 30 min [61]. 

The use of telemonitoring platforms at CPAP initiation provide an opportunity to combine lifestyle interventions and patient engagement, supported by telemedicine for integrated care in OSA. An example of holistic telerehabilitation lifestyle intervention at CPAP initiation has recently been described using the intervention mapping framework [62] and the results of this multi-center randomized controlled trial can be used to inform the design of future interventions [63]. 

Telemedicine can provide application tools to collect PROMs which can be considered as a predictor of patient’s outcome. The development of such tools allows us to collect PROMs over time and to be able to assess, with a dynamic process, the association between PROMs’ changes and patient outcomes, such as CPAP adherence or sleepiness [59,64]. Causal inference approaches can be used to analyze retrospective databases and provide real-world based evidence of the impact of advanced management tools based on telemedicine approaches on PROMs, such as CPAP adherence [65].

## 5. From Teleconsultations to Collaborative Care Models

To move towards personalized care in OSA [66], management pathways should be designed to provide a comprehensive solution that takes into account the heterogeneity of clinical phenotypes and the dynamics of lifespan trajectories of individuals with OSA. The implementation of remote management that relies on the participation of homecare providers and sleep physicians [67] offers the unique opportunity to provide holistic treatment plans that place the patient at the center of his/her care. Remote management pathways rely on virtual consultation and follow-up visits that are delivered either by telephone or videoconferencing [68]. Virtual care was shown to be as effective as in-person consultations for improving sleepiness in CPAP-treated OSA patients [69]. The authors of this meta-analysis suggest that, on the basis of the patient’s preference, remote management of patients with OSA using CPAP should be available as an alternative care strategy to in-person follow-up [69].

The telemonitoring of CPAP treatment is another important component of remote management pathways. Telemonitoring has been shown to be as effective as in-person care for improving PAP adherence [65,70] and does not imply increased costs [69]. Telemonitoring is transforming patient follow-up by being implanted in dedicated virtual platforms [50]. Enhancing patients’ participation in care through remote management pathways could facilitate the advancement of shared decision-making in the realm of OSA management [71]. 

Multimodal telemonitoring proves effective among OSA patients with increased cardiovascular risk, enhancing adherence to CPAP therapy and improving patient-centered outcomes including daytime sleepiness and quality of life [72]. However, its efficacy was not evident among patients with lower cardiovascular risk [73].

A recent randomized controlled trial in patients with OSA and obesity (168 patients recruited at 16 centers in Japan) examined the effects of the implementation of CPAP telemonitoring enhanced with body weight management tools (scales), BP measures, and a pedometer that could transmit data from devices wirelessly [74]. The group that benefited from multimodal telemonitoring exhibited a higher percentage of weight reduction (≥3%) compared to the standard PAP telemonitoring group.

Recent studies have shed light on the potential of wearable digital health technologies to transform healthcare by making behavioral and physiological patterns in daily life, outside the clinic, visible to healthcare professionals. A recent series in the New England Journal of Medicine has highlighted the value of these technologies in diabetes [75], two types of cardiovascular disease [57], and in the management of depression [56]. 

## 6. Pitfalls of Telemedicine

While showing promising potential for enhancing CPAP adherence and overall management of OSA, several challenges remain for the development of remote management pathways. A primary obstacle lies in the disparity experienced by certain populations [i.e., “The digital divide” [76]], such as the elderly, who could gain significant advantages from telemedicine services, but encounter challenges stemming from technical, cultural, and financial barriers [68]. However, in certain countries like France, adherence to CPAP therapy monitored remotely serves as a prerequisite for health insurance coverage of CPAP devices, and reimbursement rates are proportionate to adherence levels. Another area for improvement involves the absence of standardized calculation methods for CPAP indices across manufacturers [51], along with the necessity for thorough validation of devices utilized in multimodal telemonitoring of various health-related parameters [45]. Finally, numerous regulatory matters concerning data safety and healthcare regulation remain unresolved, and additional enhancements should concentrate on delineating the pertinent data for medical diagnosis and follow-up purposes [77].

## 7. Conclusions

Sleep apnea syndrome affects many people around the world, and is undoubtedly under-diagnosed. Given the heterogeneity of diagnosis and treatment, it is essential to develop personalized approaches. Telemedicine is one of the promising solutions for the years to come which could enable the development of individualized, digitized care pathways, as summarized in Figure 1. However, it will be important to ensure that digital development does not work to the detriment of patients, particularly those who are far removed from these systems. Telemedicine is a tool of the future, but it must be seen as one possible approach among others. 

## Figures and Tables

**Figure 1 jcm-13-02700-f001:**
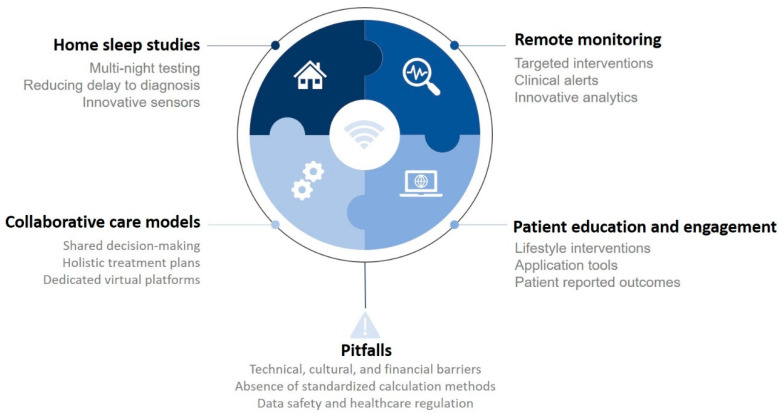
Overview of the future of telemedicine in obstructive sleep apnea.

## Data Availability

Not applicable.

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
