# Peer review of "The Future of Telemedicine for Obstructive Sleep Apnea Treatment: A Narrative Review"

_jcm, 2024, doi:10.3390/jcm13092700_

Round 1
Reviewer 1 Report
Comments and Suggestions for Authors
The authors should be congratulated for their work. The role of telemedicine-based interventions (TM) in OSAS management is still not well known. Specifically, this narrative review aimed to consolidate the evidence on access to diagnosis, increase CPAP adherence, and follow-up strategies on OSAS disease. Despite the review slightly analyzing the strengths (increasing adherence to CPAP above all) and the pitfalls of Telemedicine (digital divide, digital discrepancies in access to TeM due to cultural or economic backgrounds) concerning OSAS management, the manuscript underrepresented the potential role of telemedicine in a collaborative approach. OSAS indeed potentially leads to several comorbidities (such as LUTS, nocturia, stroke, and heart attack [ that could also be managed by telemedicine care]). Within this optic, the role of Telemedicine could be useful also to cross-identify the symptoms related to two different conditions in a differential diagnosis scheme. What is the opinion of the authors? Several authors identified a link between LUTS and OSAS for instance (PMID: 38010890). Do the authors think that the TM could represent a prompt way to guarantee OSAS patients access to urological examinations and vice versa? It would be nice to include several and more detailed clinical real-life steps that could enhance the overall value of the manuscript. Moreover, a table could be added with the main strengths and pitfalls of TM.
Author Response
The authors should be congratulated for their work. The role of telemedicine-based interventions (TM) in OSAS management is still not well known. Specifically, this narrative review aimed to consolidate the evidence on access to diagnosis, increase CPAP adherence, and follow-up strategies on OSAS disease. Despite the review slightly analyzing the strengths (increasing adherence to CPAP above all) and the pitfalls of Telemedicine (digital divide, digital discrepancies in access to TeM due to cultural or economic backgrounds) concerning OSAS management, the manuscript underrepresented the potential role of telemedicine in a collaborative approach.
SAS indeed potentially leads to several comorbidities (such as LUTS, nocturia, stroke, and heart attack [ that could also be managed by telemedicine care]).
Within this optic, the role of Telemedicine could be useful also to cross-identify the symptoms related to two different conditions in a differential diagnosis scheme.
What is the opinion of the authors?
We thank the reviewer for this comment. We have already addressed the topic of comorbidities in the two paragraphs commenting on multimodal comorbidities (page 5):
“Multimodal telemonitoring proves effective among OSA patients with increased cardiovascular risk, enhancing adherence to CPAP therapy and improving patient-centered outcomes including daytime sleepiness and quality of life [1]. However, its efficacy wasn't evident among patients with lower cardiovascular risk [2].
A recent randomized controlled trial in patients with OSA and obesity (168 patients recruited in 16 centers in Japan) examined the effects of the implementation of CPAP telemonitoring enhanced with body weight management tools (scale), BP measures and a pedometer that could transmit data from devices wirelessly [3]. The group that benefited from multimodal telemonitoring exhibited a higher percentage of weight reduction (≥3%) compared to the standard PAP telemonitoring group.”
To extend the role of telemedicine in another chronic condition, we have added and additional paragraph addressing the use of digital health wearables in various chronic diseases including depression, a common comorbidity in OSA.
“Recent studies have shed light on the potential of wearable digital health technologies to transform health care by making behavioural and physiological patterns in daily life, outside the clinic, visible to health care professionals. A recent series in the New England Journal of Medicine has highlighted the value of these technologies in diabetes [4], two types of cardiovascular disease [5] and in the management of depression [6].”
Several authors identified a link between LUTS and OSAS for instance( PMID: 38010890). Do the authors think that the TM could represent a prompt way to guarantee OSAS patients access to urological examinations and vice versa? It would be nice to include several and more detailed clinical real-life steps that could enhance the overall value of the manuscript. Moreover, a table could be added with the main strengths and pitfalls of TM.
We agree with the reviewer’s comment. Telemedicine is a promising way for OSA patients to access urological examinations and to allow more interactions between medical specialties.
Our Figure 1 summarizes the main strengths and pitfalls of telemedicine and we do not think that a table is necessary.
Reviewer 2 Report
Comments and Suggestions for Authors
Congratulations to the authors for the topic chosen for the review and how they have developed it. I have just minor suggestions to improve your paper:
- In order to improve the readability of your paper I would suggest a smoother transition between the different sections;
- to enrich your introduction and empower the idea of feasibility of telemedicine in general, I would strongly suggest to authors to include “The Feasibility, Effectiveness and Acceptance of Virtual Visits as Compared to In-Person Visits among Clinical Electrophysiology Patients during the COVID-19 Pandemic. J Clin Med. 2023 Jan 12;12(2):620. doi: 10.3390/jcm12020620. PMID: 36675547; PMCID: PMC9865180.” ;
- can you provide more studies regarding the cost-effectivness analysis?;
- can you add a table with the most relevant studies in your paper?;
Author Response
Reviewer 2
Congratulations to the authors for the topic chosen for the review and how they have developed it. I have just minor suggestions to improve your paper:
- In order to improve the readability of your paper I would suggest a smoother transition between the different sections;
We have added transitions between some of the different paragraphs to improve the readability. For example:
Page 3: “Once OSA is diagnosed, the first-line treatment for OSA is CPAP.”
Page 4: “In line with the widespread use of remote monitoring platforms, over the past few years there has been an increase in the access to “
- to enrich your introduction and empower the idea of feasibility of telemedicine in general, I would strongly suggest to authors to include “The Feasibility, Effectiveness and Acceptance of Virtual Visits as Compared to In-Person Visits among Clinical Electrophysiology Patients during the COVID-19 Pandemic. J Clin Med. 2023 Jan 12;12(2):620. doi: 10.3390/jcm12020620. PMID: 36675547; PMCID: PMC9865180.” ;
We thank the reviewer for this suggestion. We have added this reference to the introduction:
“Telemedicine has the potential to offer patients convenient access to healthcare [24] and offers remarkable potential to lower healthcare costs and enhance access to various care services for underserved populations [25]. Telemedicine is already used in various pathologies and it has been shown that it is an acceptable alternative to in-person care as evidenced in a recent study examining the feasibility, effectiveness and acceptance of virtual visits as compared to in-person visits among clinical electrophysiology patients during the Covid-19 pandemic [26].”
- can you provide more studies regarding the cost-effectivness analysis?;
We have added a study that examined the cost effectiveness of telemedicine in OSA:
“Few studies to date have investigated the cost-effectiveness of telemedicine in the management of OSA. Nevertheless, a randomized controlled trial comparing a telemedicine-based CPAP follow-up strategy compared with standard face-to-face management showed that telemedicine is cost-effective mostly due to savings on transport and less lost productivity (indirect costs) [33].”
- can you add a table with the most relevant studies in your paper?;
We do not think that such an additional table will improve the readability and the quality of the review.
- Pepin, J.L.; Jullian-Desayes, I.; Sapene, M.; Treptow, E.; Joyeux-Faure, M.; Benmerad, M.; Bailly, S.; Grillet, Y.; Stach, B.; Richard, P.; et al. Multimodal Remote Monitoring of High Cardiovascular Risk Patients With OSA Initiating CPAP: A Randomized Trial. Chest 2019, 155, 730-739, doi:10.1016/j.chest.2018.11.007.
- Tamisier, R.; Treptow, E.; Joyeux-Faure, M.; Levy, P.; Sapene, M.; Benmerad, M.; Bailly, S.; Grillet, Y.; Stach, B.; Muir, J.F.; et al. Impact of a Multimodal Telemonitoring Intervention on CPAP Adherence in Symptomatic OSA and Low Cardiovascular Risk: A Randomized Controlled Trial. Chest 2020, 158, 2136-2145, doi:10.1016/j.chest.2020.05.613.
- Murase, K.; Minami, T.; Hamada, S.; Gozal, D.; Takahashi, N.; Nakatsuka, Y.; Takeyama, H.; Tanizawa, K.; Endo, D.; Akahoshi, T.; et al. Multimodal Telemonitoring for Weight Reduction in Patients With Sleep Apnea: A Randomized Controlled Trial. Chest 2022, 162, 1373-1383, doi:10.1016/j.chest.2022.07.032.
- Hughes, M.S.; Addala, A.; Buckingham, B. Digital Technology for Diabetes. N Engl J Med 2023, 389, 2076-2086, doi:10.1056/NEJMra2215899.
- Spatz, E.S.; Ginsburg, G.S.; Rumsfeld, J.S.; Turakhia, M.P. Wearable Digital Health Technologies for Monitoring in Cardiovascular Medicine. N Engl J Med 2024, 390, 346-356, doi:10.1056/NEJMra2301903.
- Fedor, S.; Lewis, R.; Pedrelli, P.; Mischoulon, D.; Curtiss, J.; Picard, R.W. Wearable Technology in Clinical Practice for Depressive Disorder. N Engl J Med 2023, 389, 2457-2466, doi:10.1056/NEJMra2215898.
- Mariani, M.V.; Pierucci, N.; Forleo, G.B.; Schiavone, M.; Bernardini, A.; Gasperetti, A.; Mitacchione, G.; Mei, M.; Giunta, G.; Piro, A.; et al. The Feasibility, Effectiveness and Acceptance of Virtual Visits as Compared to In-Person Visits among Clinical Electrophysiology Patients during the COVID-19 Pandemic. J Clin Med 2023, 12, doi:10.3390/jcm12020620.